# Graph-Constrained Diffusion for End-to-End Path Planning

**Dingyuan Shi**
Beihang University
chnsdy@buaa.edu.cn

**Yongxin Tong**
Beihang University
yxtong@buaa.edu.cn

**Zimu Zhou**
City University of Hong Kong
zimuzhou@cityu.edu.hk

**Ke Xu**
Beihang University
kexu@buaa.edu.cn

**Zheng Wang**
Independent Researcher
wangzheng04@gmail.com

**Jieping Ye**
University of Michigan
jieping@gmail.com
jpye@umich.edu

## Abstract

Path planning underpins various applications such as transportation, logistics, and robotics. Conventionally, path planning is formulated with explicit optimization objectives such as distance or time. However, real-world data reveals that user intentions are hard-to-model, suggesting a need for data-driven path planning that implicitly incorporates the complex user intentions. In this paper, we propose GDP, a diffusion-based model for end-to-end data-driven path planning. It effectively learns path patterns via a novel diffusion process that incorporates constraints from road networks, and plans paths as conditional path generation given the origin and destination as prior evidence. GDP is the first solution that bypasses the traditional search-based frameworks, a long-standing performance bottleneck in path planning. We validate the efficacy of GDP on two real-world datasets. Our GDP beats strong baselines by $14.2\% \sim 43.5\%$ and achieves state-of-the-art performances.

## 1 Introduction

Path planning identifies an optimal sequence of vertices that forms a path between the given origin and destination in a road network graph. It underpins applications across a wide spectrum of fields, including transportation, logistics, emergency services, and robotics. Traditionally, path planning is framed as a combinatorial optimization problem with *well-defined* objectives, such as minimizing the total travel distance or time. Established *search-based* algorithms like A* or Dijkstra's algorithm Hart et al. (1968); Dijkstra (1959) are then employed to search a path on the road network graph, where each edge is assigned a specific cost that reflects the objective in question.

However, data from real-world applications reveals that many users opt for paths that are neither the shortest nor the fastest Quercia et al. (2014). This divergence suggests the complexity to model the user intentions in path planning via the conventional search-based framework under explicit objectives. Such complexity is mainly because the objectives often incorporate numerous variables that are not easily modeled in closed form. For example, users might prioritize scenic paths over shorter ones for leisure drives Ceikute & Jensen (2013). Another possible reason is that the search-based algorithms estimates the cost of a path as the summation of its composed edges, known as the *linearly accumulative cost assumption*, which does not always hold. For example, since the discharge rate of an electric vehicle battery is non-linear Szumanowski & Chang (2008), the energy cost will be different even passing the same edge, leading to different total energy cost. Hence, there is a need to re-examine the methodologies for path planning to better align with the complex, real-world objectives observed in practice.

One thread of research tries to mine user intentions or objectives by generating paths as close to those in real-world applications as possible. This has evolved from simple Markov models Baratchi et al. (2014) to advanced sequence-to-sequence neural networks Yu et al. (2017); Wang et al. (2022b). These models are adept at producing paths from learned distributions *unconditionally*. However, their

adaptability is limited in *conditional* settings,*e.g.*, path planning with given origin and destination, since they are primarily tailored to capture the inherent distribution of path datasets.

Other efforts go beyond mere pattern recognition, leveraging these identified patterns to guide path planning Tian et al. (2023); Fu & Lee (2021); Jain et al. (2021); Wang et al. (2022a). A prevalent strategy is to transform the learned patterns from path data into edge weights. These weights are then integrated into search-based algorithms like A* or Dijkstra's algorithm Wang et al. (2019); Kong et al. (2019); Jain et al. (2021); Wang et al. (2022a); Liu & Jiang (2022). These methods adopt various edge weight designs Wang et al. (2022a); Jain et al. (2021); Liu et al. (2020); Wang et al. (2019); Kong et al. (2019), yet their performance is limited by the search-based framework. Specifically, search-based algorithms assume that the path cost is linearly accumulative, which may not hold and could introduce bias, especially for long paths. A remedy is to avoid planning long paths exploiting key inter-relay vertices Tian et al. (2023); Fu & Lee (2021), but the reliance on the search-based framework remains a performance bottleneck.

In this work, we advocate an *end-to-end* approach to path planning. It underscores two aspects. *(i)* We directly plans on the road network graph, adhering to the distribution of existing path data, instead of relying on preset optimization goals that may fail to capture the complex, real-world user intentions. *(ii)* We bypass the traditional search-based algorithms, thus eliminating the probabilistic constraints of linear accumulative properties, which would also provide potential benefit for planning. Our idea is to harness generative models to *implicitly* mine the complex user intentions from large-scale historical path data, and to *directly* plan paths from the learned user intentions as a *conditional sampling* problem. Specifically, we opted for a *diffusion*-based model for end-to-end path planning for its flexibility and superior performance as generative models Saharia et al. (2022); Yi et al. (2023); Ajay et al. (2023). However, designing such an end-to-end solution is non-trivial due to the following challenges. *(i)* How to design a diffusion model for path generation under graph constraints? *(ii)* How to enable conditional generation for path planning with given origin and destination? The graph structure prohibits the adoption of traditional diffusion model in Euclidean space, and it also requires extra design for graph structure capturing.

In this work, we propose a new diffusion model, termed Graph-constrained Diffusion for Planning (GDP). It enables end-to-end path planning via conditional sampling and is also capable of unconditional path generation. Inspired by the physics of heat conduction, we devise a novel diffusion-based unconditional path generation model that explicitly accounts for the constraints imposed by road networks. We then exploit a tailored self-attention mechanism to transform the destination and other spatiotemporal information into prior evidence for conditional sampling from the path generation model. Our main contributions are threefold.

- We introduce a novel diffusion model for path generation. To the best of our knowledge, it is the first diffusion model for generating paths while complying with graph constraints imposed by road networks.
- We design a conditional sampling scheme to enable path planning via our diffusion model. It enables end-to-end data-driven path planning, which eliminates the performance bottleneck in conventional search-based frameworks.
- We validate the efficacy of our approach through experiments on two real-world datasets. Our GDP beats strong baselines by $14.2\% \sim 43.5\%$ and achieves state-of-the-art performances.

## 2 PROBLEM DEFINITION

We define a path on a road network as follows.

**Definition 1** (Path). *Consider an undirected graph $G = \langle V, E \rangle$ with vertices $V$ and edges $E$. If $u, v \in V$ is adjacent, then $(u, v) \in E$. A path $\mathbf{x}$ is defined as a sequence of vertices $(v_0, v_1, ..., v_{|\mathbf{x}|})$ where each pair of vertices are adjacent,* i.e., $\forall i = 0, 1, ..., |\mathbf{x}| - 1, (v_i, v_{i+1}) \in E$.

We assume $G$ is connected without self-loop or parallel edges, which is reasonable for a road network. Conventionally, path planning is formulated as the following *optimization* problem.

**Definition 2** (Path planning). *Given weight $w_e$ assigned to each edge and an origin-destination pair (OD pair) $\langle ori, dst \rangle$ which contains two vertices, path planning aims to find a path $\mathbf{x} = (v_0, ..., v_{|\mathbf{x}|})$ such that $v_0 = ori$ and $v_{|\mathbf{x}|} = dst$, and the path minimizes $\sum_{i=0}^{|\mathbf{x}|-1} w_{(v_i, v_{i+1})}$.*

As explained in the introduction, explicitly assigned edge weights and preset optimization objectives may fail to capture the complex user intentions in real-world path planning applications. Accordingly, we formulate path planning as the following *generation* problem.

**Definition 3** (End-to-end path planning). *Given a path dataset $\mathcal{P}$ that contains $|\mathcal{P}|$ paths, we aim to plan paths for given OD pairs, such that the planned paths follow the distribution of $\mathcal{P}$.*

We focus on generic path planning Jain et al. (2021); Tian et al. (2023) and will extend to personalized path planning by integrating user profile data in future work.

Table 1 summarizes the important notations that will be used throughout this paper.

Table 1: Notations

| NOTATIONS | DESCRIPTION |
|---|---|
| $G = \langle V, E \rangle$ | A graph and its vertices and edges |
| $\mathbf{x}, \mathcal{P}$ | A path and path dataset |
| $(v_0, v_1, ..., v_{|\mathbf{x}|})$ | A vertex sequence of a path $P$ |
| $ori, dst$ | origin and destination of path planning task |
| $\mathbf{x}^i, v_i$ | The i-th vertex of path $\mathbf{x}$ |
| $\mathbf{x}_t$ | The diffused path at time step $t$ |
| $\mathbf{Q}$ | $|V| \times |V|$ transition probability matrix |
| $\mathbf{A}, \mathbf{D}$ | Adjacent matrix and degree matrix of a graph |
| $\mathbf{M}[i, j]$ | The i-th row and j-th column element of matrix $\mathbf{M}$ |
| $\mathbf{M}[:, j]/\mathbf{M}[i, :]$ | The i-th row / j-th column of a matrix $\mathbf{M}$ |
| $\mathbf{C}_\tau$ | Transition probability matrix for diffusion process |
| $\mathbf{p}, q(\cdot)$ | Row vectors with length $|V|$ indicating a categorical distribution |
| $\mathbf{v}$ | One-hot row vector with length $|V|$ indicating a vertex |
| $\hat{\mathbf{v}}$ | Row vector indicating the estimated distribution for a vertex |
| $\text{Cat}(\cdot|\mathbf{p})$ | A categorical random variable with distribution $\mathbf{p}$ |
| $\alpha_t, \beta_t$ | Hyper parameters that control the diffusion scale |

## 3 SOLUTION OVERVIEW

We solve the end-to-end path planning problem by converting it into a conditional sampling task. Its objective is to determine the probability distribution of paths given an origin and destination, represented as $\tilde{p}(\mathbf{x})$. Then drawing a sample from this distribution yields a planned path. Following Janner et al. (2022), the conditional sampling probability can be expressed as:

$$\tilde{p}(\mathbf{x}) = p_\theta(\mathbf{x})h(\mathbf{x}|ori, dst) \qquad (1)$$

where $p_\theta(\mathbf{x})$ denotes the unconditional probability and $h(\mathbf{x}|ori, dst)$ represents the prior evidence. Consequently, our solution focuses on two designs.

- How to determine the unconditional path probability, *i.e.*, $p_\theta(\mathbf{x})$?
- How to incorporate the origin and destination data as prior evidence, *i.e.*, $h(\mathbf{x}|ori, dst)$?

We elaborate on the corresponding designs in the next two sections, respectively.

## 4 DIFFUSION-BASED UNCONDITIONAL PATH DISTRIBUTION MODELING

As mentioned in Section 3, we start with a generative model that models the path distribution, independent of the origin and destination. Despite studies Baratchi et al. (2014); Yu et al. (2017); Wu et al. (2017); Wang et al. (2022b) that explored various sequence-to-sequence models for path pattern mining, we opt for the *diffusion model* Sohl-Dickstein et al. (2015). The reasons are two-fold. *(i)* Diffusion models demonstrate superior performances in complex generation tasks Saharia et al. (2022); Ajay et al. (2023); Yi et al. (2023). *(ii)* Diffusion models are more flexible to incorporate categorical constraints such as those imposed by road networks than conventional sequence-to-sequence models.

## 4.1 Principles and Requirements for Diffusion Process Design

To generate high-quality paths, our diffusion model directly operates on road networks by explicitly incorporating the connectivity constraints imposed by the graph $G$. We draw insights from categorical diffusion models Austin et al. (2021); Hoogeboom et al. (2021), since vertices are categorical variables. The general framework is as below:

$$q(v_t|v_{t-1}) = \text{Cat}(v_t|\mathbf{p} = q(v_{t-1})\mathbf{Q}_t) \tag{2}$$

where $v_t$ denotes a categorical value at diffusion step $t$, $q(\cdot)$ is a row vector of categorical distribution and $\mathbf{Q}_t$ represents the transition probability matrix. $\text{Cat}(\cdot)$ denotes a categorical distribution. The i-th row and j-th column value of $\mathbf{Q}_t$ is the transition probability from category $i$ to $j$.

In generic categorical diffusion, a prevalent choice for $\mathbf{Q}_t$ is $\alpha_t\mathbf{I}_{|V|} + \beta_t\mathbf{1}\mathbf{1}^T/|V|$, where $\mathbf{I}_{|V|}$ is an identity matrix with $|V|$ rows and columns and $\mathbf{1}$ is a column vector with all elements set to 1 Austin et al. (2021); Hoogeboom et al. (2021); Yi et al. (2023). The hyperparameters $\alpha_t$ and $\beta_t \equiv 1 - \alpha_t$ modulate the forward diffusion process's scale. Essentially, $\mathbf{Q}_t$ slightly perturbs the original vertex's probability and uniformly redistributes it across other vertices. As $t$ goes to infinity, the distribution of $q(v_t)$ will ultimately converge to a uniform distribution.

Notably, Austin et al. (2021) highlighted the significance of designing a domain-specific transition probability matrix. Yet, a tailored transition probability matrix for graph topology remains elusive in existing literature, a void our research seeks to fill. In our graph-constrained situation, where all vertices are situated in a graph space, a generic categorical diffusion process fails to capture this structure. In fact, we hope the forward diffusion process should exhibit *locality*. This locality property will condense the probability around the original vertex with smaller diffusion step $t$, hence the process will capture the graph structure. Analogous to image generation, smaller $t$ values yield reduced covariance, keeping pixel values close to their origin. As $t$ grows, pixel values trend towards a normal distribution. Similarly, in our context, each "pixel" corresponds to a vertex, and we desire the diffusion process to reflect this locality, especially for smaller $t$. Our goal is for the diffusion process to spread probabilities across neighboring vertices, converging to uniformity as $t$ approaches infinity.

Together with commonly properties sought after in diffusion models Yi et al. (2023), the **requirements** are: *(i)* offers a closed form for forward process, *(ii)* ensures a computationally feasible posterior $q(v_{t-1}|v_t, v_0)$, *(iii)* makes $q(v_T)$ independent of $q(v_0)$ for uninformed sample generation, and *(iv)* exhibits locality for smaller $t$ values.

To address these, we employ the heat conduction partial differential equation, detailed in next sections. We first design diffusion process for vertices in graph space, then extend it to path, *i.e.*, a sequence of vertices.

## 4.2 Diffusion Process for a Single Vertex

We incorporate heat conduction process to construct our diffusion process. Specifically, we view probability as heat on graph, then use conduction process to change our probability for diffusion process.

**Heat Conduction on Graph.** Considering each vertex as a point heat source, heat diffuses uniformly across all vertices. While "diffuse" typically has a different connotation in diffusion models, in this context, it pertains to heat conduction on the graph. Henceforth, we use "heat conduction" to avoid confusion. The heat conduction equation is given by:

$$\frac{\partial \mathbf{h}}{\partial t} = \Delta \mathbf{h} \tag{3}$$

where $\mathbf{h}$ is a row vector of size $|V|$, indicating each vertex's heat. The Laplacian operator, $\Delta$, for a vertex is the sum of differences between it and its neighbors. This equation suggests a vertex's heat change rate depends on its heat difference with neighbors. Using the adjacency matrix $\mathbf{A}$ and degree matrix $\mathbf{D}$, we can express this as $\frac{\partial \mathbf{h}}{\partial t} = \mathbf{h}(\mathbf{A} - \mathbf{D})$, where $\mathbf{A}$ is the adjacency matrix and $\mathbf{D}$ is a diagonal matrix with $\mathbf{D}[i, i]$ representing a vertex's degree. Solving this equation yields Chamberlain et al. (2021):

$$\mathbf{h}_t = \mathbf{h}_0 \mathbf{C}_t \tag{4}$$

with transition matrix $\mathbf{C}_t = e^{(\mathbf{A}-\mathbf{D})t}$. This represents heat conduction on a graph over time $t$.

**Forward Diffusion Process.** By substituting the heat of each vertex with probabilities in Equation (4), we can derive our diffusion process as follows

$$q(v_t|v_{t-1}) = \text{Cat}(v_t|q(v_{t-1})\mathbf{C}_\tau) \tag{5}$$

The matrix $\mathbf{C}_\tau$ has several beneficial **properties**: *(i)* $\mathbf{C}_\tau$ is a symmetric matrix *i.e.*, $\mathbf{C}_\tau^T = \mathbf{C}_\tau$. *(ii)* $\mathbf{C}_{\tau_1+\tau_2} = \mathbf{C}_{\tau_1}\mathbf{C}_{\tau_2}$, $\mathbf{C}_{k\tau} = (\mathbf{C}_\tau)^k$. *(iii)* $\mathbf{C}_\tau \to \mathbf{1}\mathbf{1}^T/|V|$ as $\tau \to \infty$ if graph is connected. *(iv)* $\mathbf{C}_\tau \to \mathbf{I}$ as $\tau \to 0$. *(v)* Summation of each row or column for $\mathbf{C}_\tau$ is $1 \; \forall \tau > 0$, if graph is connected. The proof for these properties is provided in (App. A).

By incorporating a series of hyper parameters $\beta_t$, we can derive the closed form of forward process

$$q(v_t|v_0) = \text{Cat}(v_t|\mathbf{p} = \mathbf{v}_0\bar{C}_t) \tag{6}$$

where $\bar{C}_t = C_1 C_2 ... C_t = C_{\beta_1}...C_{\beta_t} = C_{\sum_{i=1}^t \beta_i}$. $\mathbf{v}_0$ is a one-hot row vector indicating the vertex.

Note that property *(i)* simplifies notation by omitting the transpose superscript. Property *(ii)* provides a closed form for the forward process (see Eq.(6)), which fulfills requirement *(i)*. Property *(iii)* satisfies requirement *(iii)* because it ensures the final probability $q(v_T)$ is a uniform distribution independent of $q(v_0)$. Property *(iv)* captures the desired locality (requirement *(iv)*). Property *(v)* guarantees $p(v_t)$ remains a valid probability distribution since the sum of $\mathbf{v}$ is 1. A visual representation of probability diffusion on a graph is shown in Figure 1, where probability initially centered on one vertex disperses across others over time.

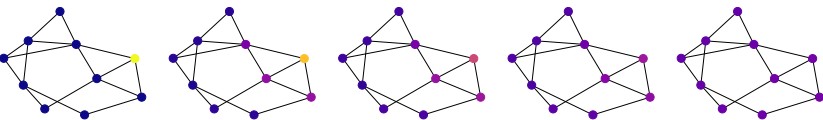

Figure 1: Illustration for forward process of a single vertex. Warmer color means larger probabilities.

Our forward process meet the requirements *(i)*,*(ii)* and *(iv)* as we have mentioned above. Next we design the reverse process to meet requirement *(ii)*. *i.e.*, the posterior $q(v_{t-1}|v_t, v_0)$ is tractable.

**Reverse Process.** To derive the reverse process, our goal is to find a tractable posterior probability distribution $q(v_{t-1}|v_t, v_0)$. Utilizing the Bayesian rule, the Markov property and symmetry property (property *(i)*) of the diffusion process, we have $q(v_{t-1}|v_t, \hat{v}_0) = \text{Cat}(v_{t-1}|\mathbf{p} \propto \mathbf{v}_t\mathbf{C}_t \odot \mathbf{v}_0\bar{\mathbf{C}}_{t-1})$. where $\odot$ denotes element-wise multiplication. When calculating the posterior $q(v_{t-1}|v_t, v_0)$, however, we do not have the original $\mathbf{v}_0$. Hence we can only use the estimated $\hat{\mathbf{v}}_0$ which is the estimated distributions of $v_0$. Thus we have

$$q(v_{t-1}|v_t, \hat{v}_0) = \text{Cat}(v_{t-1}|\mathbf{p} \propto \mathbf{v}_t\mathbf{C}_t \odot \hat{\mathbf{v}}_0\bar{\mathbf{C}}_{t-1}) \tag{7}$$

The detailed derivation is provided in (App. B).

### 4.3 DIFFUSION PROCESS FOR A PATH

We have detailed the diffusion process for individual vertices. Now, we address paths, sequences of vertices. We differentiate between "time" in diffusion steps (subscripts) and vertex order in paths (superscripts). For instance, $\mathbf{x}_t^i$ denotes the i-th vertex in path $\mathbf{x}$ at diffusion step $t$. At $t = 0$, $\mathbf{x}_0 = \mathbf{x}$ is the initial path.

For path diffusion, we can either strictly maintain vertex connectivity or treat it as contextual. We opt against strict connectivity for two reasons. *(i)* Preserving connectivity without information loss is complex. It introduces conditional probabilities like $p(\mathbf{x}_t^i|\mathbf{x}_t^{i-1})$, complicating reverse processes and conflicting with diffusion models' non auto-regressive nature Janner et al. (2022). *(iii)* In real road networks, many vertices have limited adjacency. Strict connectivity restricts diffusion space, making $q(\mathbf{x}_T)$ overly reliant on $q(\mathbf{x}_0)$, violating requirement *(ii)*.

Thus, we diffuse each vertex independently. While individual vertices retain their locality in paths, the entire path's vertices stay close to their original positions for small $t$. For large $t$, vertices become random, aiding sample generation.

The forward closed form process therefore can be represented as

$$q(\mathbf{x}_t|\mathbf{x}_0) = \Pi_{i=1}^{|\mathbf{x}|} q(\mathbf{x}_t^i|\mathbf{x}_0^i) = \otimes_{i=1}^{|\mathbf{x}|} \mathrm{Cat}(\mathbf{x}_t^i|\mathbf{p} = \mathbf{x}_0^i \bar{\mathbf{C}}_t) \qquad (8)$$

As for reverse process, it can be represented as follows.

$$q(\mathbf{x}_{t-1}|\mathbf{x}_t, \hat{\mathbf{x}}_0) = \otimes_{i=1}^{|\mathbf{x}|} \mathrm{Cat}(\mathbf{x}_{t-1}^i|\mathbf{p} \propto \mathbf{x}_t^i \mathbf{C}_t \odot \hat{\mathbf{x}}_0^i \bar{\mathbf{C}}_{t-1}) \qquad (9)$$

where $\otimes$ indicates Cartesian product and $|\mathbf{x}|$ denotes the lengths of path $\mathbf{x}$.

Note that in reverse process, each vertex distribution is not independent with each others. This is because when estimating $\hat{\mathbf{x}}_0$, we build an neural network $nn_\theta$. It takes the $\mathbf{x}_t$ and diffusion time step $t$ as input (*i.e.*, $nn_\theta(\mathbf{x}_t, t) = p_\theta(\hat{\mathbf{x}}_0|\mathbf{x}_t)$), then output the estimate of $q(\mathbf{x}_{t-1}|\mathbf{x}_t, \hat{\mathbf{x}}_0)$ by Eq.(9). This $nn_\theta$ adopts a U-net structure Ronneberger et al. (2015), and its convolution together with fully connected layers help it model dependency among vertices. The detailed design is shown in (App. C.1).

To ensure the final path connectivity, we do not directly sample on $\hat{\mathbf{x}}_0$. Instead, we view $\hat{\mathbf{x}}_0$ as a proposal and conduct a beam search. The detailed sampling process can be found at (App. C.2).

## 5 PATH PLANNING AS CONDITIONAL SAMPLING

As mentioned in Section 3, we are required to devise a model to calculate $h(\mathbf{x}|ori, dst)$. A trivial method is simply adding the source and destination vertex information as condition to guide the generation process. However, this method can hardly work, mainly because path planning has a strong spatial property. So we need to build features for spatial property for better generation. Specifically, we need to *(i)* build spatial features as prior evidence and *(ii)* integrate the prior evidence into our unconditional sampling.

**Make Spatial Features as Prior Evidence.** To build spatial features as prior evidence, we design an attention-based model $seq_\phi$ (see Figure 5). It takes the origin, destination and prefix of the current path as input, then outputs the probability of the next vertex. Please refer to (App. D) for detailed structure.

**Sampling Process.** Together with the diffusion model, we can now conduct path planning as conditional generation (see Algorithm 3 in (App. E)). The overall steps can be summarized as below: it *(i)* gets the next vertex probability based on prefix path (which contains only origins initially) by Eq.(1) based on model $nn_\theta$ and $seq_\phi$. And *(ii)* samples from the probability to get the next vertex and *(iii)* appends it to the prefix for next prediction till we hit the destination or reach the maximum length restriction. The core of the above steps is the calculation of Eq.(1). The prior evidence term $h(\cdot|ori, dst)$ can be calculated based on $seq_\phi$. While for the unconditional probability term $p(\cdot)$, we need to bring in the current prefix information for generation. So we first diffuse the prefix by Eq.(8) and concatenate with a uniformly sampled sequence, then our diffusion model $nn_\theta$ conduct reverse process to denoise the whole sequence, hence get the estimated $\hat{\mathbf{x}}_0$. To avoid frequent call of the forward and reverse diffusion process, we double the length of uniformly sampled sequence for concatenation in each iteration, hence the length of planned path will increase exponentially.

## 6 EXPERIMENTS

### 6.1 EXPERIMENTAL SETTING

**Dataset.** We use two real datasets city A and city B from Didi Gaia [1]. Please refer to (App. F.1) for preprocess, parameter setting and other implementation detail.

**Baselines.** For path planning, we choose four below algorithms as our baselines. **Dijkstra's algorithm (DA)** Dijkstra (1959) searches the shortest distance path for given OD pairs. **NMLR** Jain et al. (2021) learns the distributions based on Markov property assumption and search the paths with the largest probabilities each step. **Key Segment (KS)** Tian et al. (2023) detects the relay vertex for

---

[1] https://gaia.didichuxing.com

the given OD pairs, then plan paths from origin to relay and then to the destination. **Navigation API from Amap (Navi)** calls the planning API from Amap [2] to get the path.

Also, we compare with path generation algorithms to directly justify the effectiveness of our generative model. **N-gram** directly estimates the transition probability $p(v_t|v_{t-1}, ..., v_{t-n+1})$ by counting the frequency. **HMM** Yin et al. (2018) is a hidden Markov model based algorithm which reduces state numbers. **CSSRNN** Wu et al. (2017) and **MTNet** Wang et al. (2022b) both adopt an LSTM-based model for better capturing long sequence pattern.

**Evaluation Metrics.** For planning, we use Dynamic Time Wrapping (DTW) Müller (2007) and Longest Common Subsequence (LCS)Bergroth et al. (2000) to measure the similarity between planned paths and the ground truth paths. A smaller DTW or a larger LCS indicates better performance.

For generation, the most commonly used metrics is Neg-Log-Likelihood (NLL) Wu et al. (2017); Wang et al. (2022b). Apart from that, we also use Kullback-Leibler divergence w.r.t. Edge Visit distribution (KLEV). KLEV indicates a first-order transition similarity between two datasets. By replacing Kullback-Leibler divergence with Jensen-Shannon divergence, we analogously define JSEV. NLL and KLEV/JSEV evaluate the performance from perspective of high-order and first-order transition, respectively. The specific definition can be found in (App. F.1).

## 6.2 EVALUATION RESULTS

Table 2: Evaluations for path planning. Smaller DTW or larger LCS means better performance.

| City | Metrics | Methods | | | | |
|---|---|---|---|---|---|---|
| | | DA | NMLR | KS | Navi | GDP (Ours) |
| A | DTW | — | 274.2% | −1.2% | −11.1% | **−30.9%** |
| | LCS | — | 5.59% | 24.1% | 45.9% | **80.8%** |
| B | DTW | — | 250.8% | 1.8% | −3.16% | **−15.9%** |
| | LCS | — | 9.02% | 32.2% | 55.9% | **82.6%** |

**Path planning evaluation.** For path planning evaluation, we sample 1000 number of paths uniformly from test datasets. Then we conduct path planning and compare the similarities(See Table 2). For convenience we take Dijkstra's algorithm as a benchmark, and the performance of other algorithms is represented by the percentage improvement relative to Dijkstra's algorithm. Please refer (App. F.2) to for the raw results. Our GDP performs best among all other baselines (*e.g.*, improves LCS by 80.8% and reduces DTW by 30.9% with regard to Dijkstra's method in City A). We notice that, the most compatible baseline is Navi, which indicating that the algorithm deployed in real industry also adopts data-driven approaches. Also, the most recent existing method KS achieves second best in baselines. We also find some existing methods do not out perform naive Dijkstra's methods well. This is probably because NMLR still adopts the Markov Property which confines the capture of higher order relationship underlying path data.

Another important question is how our method can ensure to hit the destination. We use both test and shuffled OD pairs for validation. The test OD pairs are directly sampled from the test dataset our model has never seen. To generate the shuffled OD pairs, we first sample OD pairs from the test dataset and then shuffle them. Shuffle operation adds more difficulty to our model for planning tasks, since the shuffled OD pair distribution is different from the original test dataset, forcing our model to use sub-level patterns for generalization. Overall, our method keeps the hit ratio from 94.2% to 99% average, indicating its availability. Please refer to (App. F.3) for detailed discussion.

Next we present some cases of City A for illustration. Again, we use test and shuffled OD pairs. For test OD pairs, we use test dataset as ground truth. For shuffled OD pairs, we can not get ground truth from dataset so we call the Navi for comparison. Figure 2 provides illustrations of the planned paths look like.

**Unconditional generation evaluation.** The generation comparison among all methods with respect to all metrics are shown in Table 3. Our GDP algorithm outperforms all baselines overall datasets. For NLL, it is shown that neural network based model (CSSRNN, MTNet and our GDP) perform significantly better than count based model (N-gram, HMM). This is because NLL evaluates high order

---

[2]https://lbs.amap.com

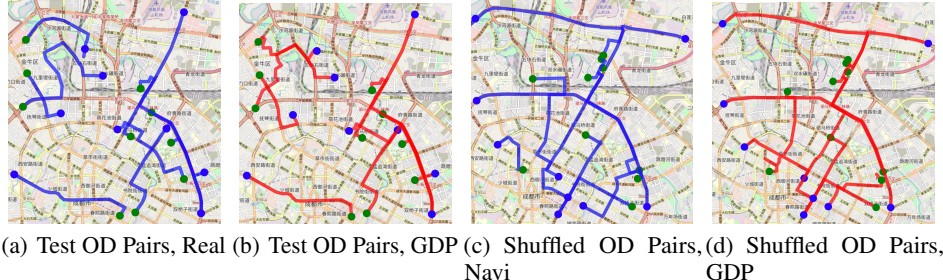

(a) Test OD Pairs, Real (b) Test OD Pairs, GDP (c) Shuffled OD Pairs, Navi (d) Shuffled OD Pairs, GDP

Figure 2: Blue/red lines indicates the ground truth/planned paths.

Table 3: Evaluations for unconditional generation. Lower values means better performance.

| City | Metrics | Methods | | | | | |
|------|---------|---------|-----|--------|-------|-----------|------|
| | | N-gram | HMM | CSSRNN | MTNet | GDP (Ours) | GDP- |
| A | NLL | 300.144 | 278.238 | 27.742 | 69.859 | **24.697** | 43.582 |
| | KLEV | 7.300 | 6.804 | 7.348 | 6.274 | **5.724** | 7.273 |
| | JSEV | 3.309 | 3.061 | 3.3325 | 2.795 | **2.518** | 3.295 |
| B | NLL | 319.662 | 302.506 | 27.890 | 83.679 | **23.740** | 47.418 |
| | KLEV | 7.719 | 6.863 | 7.356 | 6.419 | **6.128** | 7.678 |
| | JSEV | 3.521 | 3.092 | 3.337 | 2.869 | **2.724** | 3.500 |

transition probability and nerual network models are designed to capture this sequence information. Those count-based models, however, are based on Markov property assumption, can hardly capture high order information, leading to a poor performance. As for first order evaluation metrics (*i.e.*, KLEV and JSEV), however, neural network models only slightly outperform count based baselines. This indicates that our models can capture the long sequence information as those sequence model, which is mainly attributed to the attention module of temporal U-net. For ablation study, we adopt GDP-, which is built by replacing our heat conduction diffusion method with a generic categorical diffusion (with a transition probability matrix $\mathbf{Q}_t = \alpha_t \mathbf{I}_{|V|} + \beta_t \mathbf{1}\mathbf{1}^T/|V|$). The degradation of performances testify the effectiveness of our diffusion method. Other comparison between generated and real paths can be shown in (App. F.4).

**Forward and reverse process.** Now we provide a case study to illustrate how our diffusion process is working (See figure 3). We randomly pick 5000 paths from city B. In the forward process, each vertex is gradually perturbed to other vertices. Because of the locality property of our process, when $t$ is small the vertices tend to stay close to original vertices. As $t$ increases, all vertices tend to uniformly distributed, hence the paths become out of order. In the reverse process, the estimated $\hat{\mathbf{x}}_0$ gradually reverse back to the original paths.

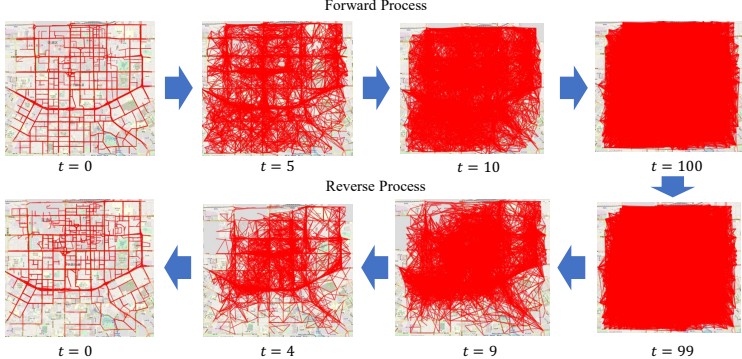

Figure 3: The forward and reverse process for paths in city B

Other experiments such as execution efficiency are put into (App. F) due to the page limit.

## 7 RELATED WORK

**Path Generation for Pattern Mining.** Path pattern mining dates back to the 1990s, where locations are modeled as states in Markov chain models Sutton & Barto (1998). Hierarchical hidden states can

be used to mitigate the sparsity in the vast state space Baratchi et al. (2014), while contextual data *e.g.*, weather can be integrated to enhance path generation accuracy Yin et al. (2018).

To capture complex patterns, the recent approach is to combine Markov Decision Process (MDP) with reinforcement learning. For example, location transitions can be exploited as routing actions to optimize the MDP via value iteration Zheng & Ni (2014). A Generative Adversarial Network (GAN) can be integrated as the reward function for further refinement Yu et al. (2017); Choi et al. (2021).

To model long paths, sequence-to-sequence (seq2seq) models have been employed. As seq2seq models are primarily devised for language processing, the challenge is to comply with the topological constraints of road networks. Some studies introduced a "lookup" operation, enabling recurrent neural networks for path generation Wu et al. (2017). Others masked the final output of the probability distribution to prevent invalid vertices Wang et al. (2022b). The status quo adopts an encoder-decoder structure: encode the path sequence into a latent space, train a generative model on the latent vectors, and decode the latent vectors back to paths Rao et al. (2021); Huang et al. (2019); Feng et al. (2020).

Despite extensive path generation studies for pattern mining, they generate paths *unconditionally*, and are not easily extendable to conditional generation, thus limiting their applicability to path planning.

**Data-driven Path Planning.** Data-driven approaches can be employed to capture the hard-to-model user intentions to improve path planning Quercia et al. (2014). Due to the difficulty to ensuring connectivity and reaching the destination in a pure data-driven solution, existing solutions halt at incorporating data-driven insights with traditional search-based algorithms. They differ in the edge weight designs, *e.g.*, negative log-likelihood Wang et al. (2022a); Jain et al. (2021), heuristics from inverse reinforcement learning Liu et al. (2020), and neural network informed weights Wang et al. (2019); Kong et al. (2019). Yet the search-based framework fundamentally limits the ability to capture the non-linear, high-order dependencies in paths, especially in long paths. One remedy is to identify key inter-relay vertices to reduce the path lengths in planning Tian et al. (2023); Fu & Lee (2021). However, the search-based framework remains the primary performance bottleneck.

Our work falls into this thread of research, but completely breaks free from the search-based framework, thus enabling an *end-to-end* data-driven path planning solution.

**Categorical Diffusion Models.** Diffusion models Sohl-Dickstein et al. (2015) prove effective in complex generation tasks across various domains, including computer vision Saharia et al. (2022), chemistry Yi et al. (2023), and robotics planning Ajay et al. (2023). Many interesting diffusion-based planner such as Carvalho et al. (2023)) happens in Euclidean space and are not suitable in our graph space. Our work is inspired by the advances in categorical diffusion models Austin et al. (2021); Hoogeboom et al. (2021), where they enables diffusion process for categorical values. However, they do not easily adapt to graph constraints.

Our work is most related to Zhu et al. (2023), where a diffusion model is applied to generate GPS-coordinate trajectories. Since the GPS coordinates represent numerical values in a two-dimensional Euclidean space rather than graph space. In planning task, however, we are required to output a path on road network graph since the post-process of navigation will integrate other information like lanes, traffic lights and speed limitation into the path. These information is usually organized by road segments (rather than coordinates) Wang et al. (2018) and a path on graph will help easily fetch them.

To the best of our knowledge, we are the first to generate paths with explicit graph constraints using diffusion models.

## 8   CONCLUSION

This paper introduces GDP, a novel diffusion-based model tailored for end-to-end path planning. We conceptualize the path planning problem as a conditional sampling task. To effectively capture the graph topology structure, we designed a unique diffusion process. Furthermore, by employing a self-attention mechanism, we integrate OD pair information as conditions into the probability space, bypassing the traditional search framework. Evaluations demonstrate GDP's proficiency in discerning the distributions inherent in the provided path dataset. Moreover, it can plan paths whose distributions align closely with existing paths. We envision our approach will facilitate the path planning smarter for modern intelligent transportation system. Besides, the post process for path legality will slightly change the sampling distributions provided by our diffusion model. Designing an algorithm without any post process is interesting and remains an open problem.

## ACKNOWLEDGMENTS

We would like to thank the anonymous reviewers for their suggestions. This work was partially supported by National Science Foundation of China (NSFC) (Grant Nos. U21A20516, 6233000216) and Beijing Natural Science Foundation (Z230001), the Basic Research Funding in Beihang University No.YWF-22-L-531, and Didi Collaborative Research Program NO2231122-00047. Zimu Zhou's research is supported by Chow Sang Sang Group Research Fund No. 9229139. Yongxin Tong is the corresponding author in this paper.

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

APPENDIX

## A  PROOF FOR PROPERTIES OF MATRIX $\mathbf{C}_\tau$

*(i)* $\mathbf{C}_\tau$ is a symmetric matrix *i.e.*, $\mathbf{C}_\tau^T = \mathbf{C}_\tau$.

**Proof.** we have $\mathbf{C}_\tau = e^{(\mathbf{A}-\mathbf{D})\tau} = \mathbf{I} + \sum_{k=1}^{\infty} \frac{(\mathbf{A}-\mathbf{D})^k \tau^k}{k!}$. Because both $\mathbf{A}$ and $\mathbf{D}$ is symmetric, so $(\mathbf{A}-\mathbf{D})^k$ is symmetric for any $k$. Thus we have $\mathbf{C}_\tau$ is symmetric.

*(ii)* $\mathbf{C}_{\tau_1+\tau_2} = \mathbf{C}_{\tau_1}\mathbf{C}_{\tau_2}$, $\mathbf{C}_{k\tau} = (\mathbf{C}_\tau)^k$.

**Proof.** By letting $X = (\mathbf{A}-\mathbf{D})\tau_1, Y = (\mathbf{A}-\mathbf{D})\tau_2$, we have

$$
\begin{aligned}
\mathbf{C}_{\tau_1+\tau_2} = e^{X+Y} &= \sum_{n=0}^{\infty} \frac{(X+Y)^n}{n!} = \sum_{n=0}^{\infty}\sum_{k=0}^{n} \frac{C_n^k X^k Y}{n!} \\
&= \sum_{n=0}^{\infty}\sum_{k=0}^{n} \frac{X^k Y^{n-k}}{k!(n-k)!}
\end{aligned}
\tag{10}
$$

We also have

$$
\begin{aligned}
\mathbf{C}_{\tau_1}\mathbf{C}_{\tau_2} = e^X e^Y &= (\sum_{n=0}^{\infty} \frac{X^n}{n!})(\sum_{n=0}^{\infty} \frac{Y^n}{n!}) \\
&= (I + X + ...)(I + Y + ...) = I + (X+Y) + (\frac{X^2}{2!} + XY + YX + \frac{Y^2}{2!}) + ... \\
&= \sum_{n=0}^{\infty}\sum_{k=0}^{n} \frac{X^k Y^{n-k}}{k!(n-k)!}
\end{aligned}
\tag{11}
$$

where in the second line we rearrange all terms based on there orders, and the coefficient is same as the binomial coefficient. By letting $\tau_1 = \tau_2 = \tau$ and mathematical induction, we have $\mathbf{C}_{k\tau} = (\mathbf{C}_\tau)^k$.

*(iii)* $\mathbf{C}_\tau \rightarrow \mathbf{11}^T/|V|$ as $\tau \rightarrow \infty$ if graph is connected.

**Proof.** We use the heat conduction equation for proof. From the physical process, the heat will eventually evenly distributed to all vertex, so it will become $\mathbf{11}^T/|V|$ whatever the initial values.

*(iv)* $\mathbf{C}_\tau \rightarrow \mathbf{I}$ as $\tau \rightarrow 0$.

**Proof.** When $\tau \rightarrow 0$, due to the continuity, $\mathbf{C}_\tau \rightarrow \mathbf{C}_0$, *i.e.*, $\mathbf{I}$.

*(v)* Summation of each row or column for $\mathbf{C}_\tau$ is 1 $\forall \tau > 0$, if graph is connected.

**Proof.** Note that based on the physical meaning, we have $\mathbf{1}^T = \mathbf{1}^T C_\tau, \forall \tau > 0$. Thus, for each column $j$, we have $\mathbf{1}^T[j] = \mathbf{1}^T C_\tau[:, j] \Leftrightarrow \sum_i C_\tau[i,j] = 1, \forall j$. Based on the symmetric property, we get the summation for each row or column is always 1.

## B  DERIVATION FOR REVERSE PROCESS

Utilizing the Bayesian rule and the Markov property of the diffusion process, we have

$$
\begin{aligned}
q(v_{t-1}|v_t, v_0) &= q(v_t|v_{t-1}, v_0)\frac{q(v_{t-1}|v_0)}{q(v_t|v_0)} = q(v_t|v_{t-1})\frac{q(v_{t-1}|v_0)}{q(v_t|v_0)} \\
&\propto q(v_t|v_{t-1})q(v_{t-1}|v_0) \\
&= \text{Cat}(v_{t-1}|\mathbf{p} \propto \mathbf{v}_t\mathbf{C}_t \odot \mathbf{v}_0\bar{\mathbf{C}}_{t-1})
\end{aligned}
\tag{12}
$$

where $\odot$ denotes element-wise multiplication. Note that the original $q(v_t|v_{t-1})$ for any category value $k$ should be $\mathbf{v}_t\mathbf{C}_\tau^T$, where $v_t$ is one-hot row vector with $v_t[k] = 1$. This is because $\mathbf{C}_\tau[i,j]$ is the probability from vertex $i$ to $j$ and $\mathbf{v}_t\mathbf{C}_\tau^T[i,j]$ will be the probability *into* category $k$. We leverage the symmetry property (property *(i)*) and omit the transpose for convenience. When calculating

the posterior $q(v_{t-1}|v_t, v_0)$, however, we do not have the original $\mathbf{v}_0$, hence we can only use the estimated $\hat{\mathbf{v}}_0$ which is the estimated distributions of $v_0$. Thus we have

$$q(v_{t-1}|v_t, \hat{v}_0) = \text{Cat}(v_{t-1}|\mathbf{p} \propto \mathbf{v}_t\mathbf{C}_t \odot \hat{\mathbf{v}}_0\bar{\mathbf{C}}_{t-1}) \tag{13}$$

## C    DIFFUSION MODEL DESIGN

### C.1    MODEL STRUCTURE

Instead of a 2-D convolutional neural network (CNN), we employ a 1-D CNN for the temporal dimension. We have omitted the down-sampling and up-sampling blocks from U-net, as they risk significant information loss. Each path vertex is crucial, holding more significance than an image pixel. Down-sampling and up-sampling can also misalign horizons, and typical solutions like linear interpolation are not fit for categorical path data. We use the Node2Vec algorithm Grover & Leskovec (2016) to obtain pre-trained vertex embeddings, which feed into our neural network. Our architecture is depicted in Figure 4.

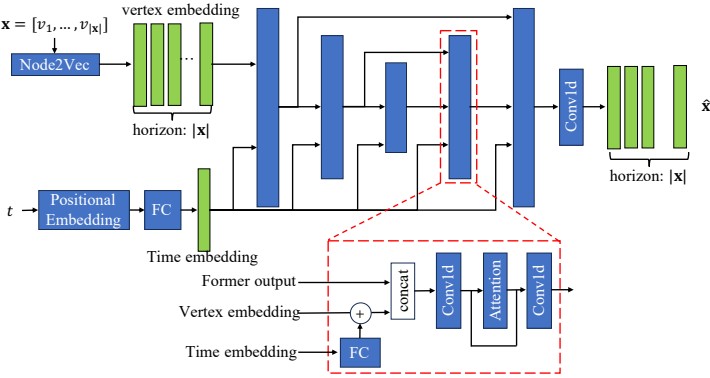

Figure 4: Our neural network design.

We use Kullback-Leibler divergence between $p_\theta(\mathbf{x}_{t-1}|\mathbf{x}_t, \hat{\mathbf{x}}_0)$ and $q(\mathbf{x}_{t-1}|\mathbf{x}_t, \mathbf{x}_0)$ as our training loss function.

Our training algorithm is shown in Algorithm 1. Next we discuss how to use our trained model for path generation.

---

**Algorithm 1:** Training

**Input:** Dataset $\mathcal{P}$, model $nn_\theta$
**Output:** model $nn_\theta$

1 **for** *training_steps* $\leftarrow 1, 2, ...$ **do**
2      $\mathbf{x}_0 \sim p_{data}(\mathbf{x}_0)$
3      $t \sim \mathcal{U}[1, T]$
4      $\mathbf{x}_t \sim q(\mathbf{x}_t|\mathbf{x}_0)$ by Eq.(8)
5      $p_\theta(\hat{\mathbf{x}}_0|\mathbf{x}_t) = nn_\theta(\mathbf{x}_t, t)$
6      calculate loss and update $nn_\theta$
7 **end**
8 **return** $nn_\theta$

---

**Algorithm 2:** Sampling

**Input:** $nn_\theta$
**Output:** A generated path $\mathbf{x}$.

1 Sample length $l$ from Gaussian mixture model
2 $\mathbf{x}_T \sim \otimes_1^l \mathcal{U}[1, |V|]$
3 **for** $t \leftarrow T, ..., 1$ **do**
4      $\hat{\mathbf{x}}_0 \leftarrow nn_\theta(\mathbf{x}_t, t)$
5      sample $\mathbf{x}_{t-1}$ by Eq.(9)
6 **end**
7 $\mathbf{x}_0 \leftarrow$ apply beam search via $\hat{\mathbf{x}}_0$
8 **return** $\mathbf{x}_0$

---

**Loss function design.**    In general, we are required to estimate $p_\theta(\mathbf{x}_{t-1}|\mathbf{x}_t, \hat{\mathbf{x}}_0) = q(\mathbf{x}_{t-1}|\mathbf{x}_t, \hat{\mathbf{x}}_0)p_\theta(\hat{\mathbf{x}}_0|\mathbf{x}_t)$.    We can minimize the Kullback-Leibler divergence between $p_\theta(\mathbf{x}_{t-1}|\mathbf{x}_t, \hat{\mathbf{x}}_0)$ and $q(\mathbf{x}_{t-1}|\mathbf{x}_t, \mathbf{x}_0)$, so we have

$$\mathcal{L}_{KL} = \mathbb{E}_{\mathbf{x_0} \sim q(\mathbf{x}_0), t \sim \mathcal{U}[1,T]}[D_{KL}(q(\mathbf{x}_{t-1}|\mathbf{x}_t, \mathbf{x}_0)||p_\theta(\mathbf{x}_{t-1}|\mathbf{x}_t, \hat{\mathbf{x}}_0))] \tag{14}$$

Some existing work Austin et al. (2021) also adds an auxiliary cross entropy loss for training.

$$\mathcal{L}_{CE} = \mathbb{E}_{\mathbf{x_0} \sim q(\mathbf{x}_0), t \sim \mathcal{U}[1,T]}[CE[\mathbf{x}_0, p_\theta(\hat{\mathbf{x}}_0|\mathbf{x}_t)]] \tag{15}$$

Apart from that, we need the generated path to be connected, so we design a loss term to punish disconnected path as below

$$\mathcal{L}_{CN} = -\mathbb{E}_{\mathbf{x_0} \sim q(\mathbf{x}_0), t \sim \mathcal{U}[1,T]}[\sum_{i=1}^{|\mathbf{x}|-1} \sum_{\hat{v}_0^i \in V} \sum_{(\hat{v}_0^i, \hat{v}_0^{i+1}) \in E} p_\theta(\hat{v}_0^i|\mathbf{x}_t) \log p_\theta(\hat{v}_0^{i+1}|\mathbf{x}_t)] \tag{16}$$

For the i-th vertex, the estimated probability is $p_\theta(\hat{v}_0^i|\mathbf{x}_0)$. If the probability is large for some vertex at order $i$, we hope the model concentrates the probabilities of next (i+1)-th vertex around the neighboring vertices of $\hat{v}_0^i$, *i.e.*, $\sum_{(\hat{v}_0^i, \hat{v}_0^{i+1}) \in E} p_\theta(\hat{v}_0^i|\mathbf{x}_t) \log p_\theta(\hat{v}_0^{i+1}|\mathbf{x}_t)$.

Therefore, our total loss can be defined as

$$\mathcal{L} = \mathcal{L}_{KL} + \lambda \mathcal{L}_{CE} + \mu \mathcal{L}_{CN} \tag{17}$$

## C.2 UNCONDITIONAL SAMPLING FOR PATH GENERATION

Our model $nn_\theta$ can take a random sampled vertex sequence as input and get an estimated denoised vertices distributions $\hat{\mathbf{x}}_0$, based on which we can sample a path. There are two issues we need to handle before utilizing the estimated $\hat{\mathbf{x}}_0$ for path sampling: *(i)* How to specified the path sequence length? *(ii)* How to maintain the path connectivity? To handle the first issue, we build an extra Gaussian mixture model. This is a simple model which models the path length distribution. In practice, a Gaussian mixture model with 4 components can model the length distribution well (see Figure 8 in (App. F.4)). To handle the second issue, we adopt a beam search. Specifically, the beam search is conducted after the reverse process when we have obtained the final estimated distribution $\hat{\mathbf{x}}_0$. Ideally, beam search operates in a breadth-first search manner. We start by sampling $n$ vertices from the distribution of $\hat{\mathbf{x}}_0^0$ (with replacement). Subsequently, we sample (up to $n$) neighbors of each vertex from the distribution $\hat{\mathbf{x}}_0^1$. This process continues, and we always keep (up to) $n$ sequences with the highest probability in a queue. Finally, we sample from these $n$ sequences to get the final path. In practice, however, setting $n$ to 1 suffices for good performance and efficient execution.

After handling the above two issues, the whole unconditional sampling process can be summarized as follows (See Algorithm 2): *(i)* Sample from the Gaussian mixture model to determine the path length (denoted as $l$); *(ii)* Generate a random vertex sequence of length $l$; *(iii)* Perform the reverse process to denoise the random vertex sequence, obtaining the estimated distribution of $\hat{\mathbf{x}}_0$; and *(iv)* Apply beam search to ensure connectivity.

## D PRIOR EVIDENCE MODEL DESIGN

Specifically, we first convert the origins and destinations into embeddings via Node2Vec Grover & Leskovec (2016), then apply a multi-head self-attention to capture the sequence information. Besides, since the planning process has a strong spatial property, we add geo-coordinations as the vertices' properties, based on which we build two extra spatial features: distance feature and direction feature. For distance feature, we calculate the distance from the current vertex to the destination based on coordinates. For direction feature, we calculate the cosine of angles between directions towards destination from the current vertex and from its adjacent vertices. Both two features will be fed into a fully connected layer and the output will be concatenated together with the output of self-attention blocks. Another fully connected layer will take all hidden features as input and output the probabilities for the next vertex. We simply adopt a cross-entropy loss to train our model and it works well.

After get the prior evidence $h(\mathbf{x}|ori, dst) = seq_\phi(ori, dst)$, we add the prior evidence into our conditional sampling by Eq.(1).

## E PSEUDOCODE FOR PLANNING ALGORITHM

The Pseudocode for our planning algorithm is shown in algorithm 3.

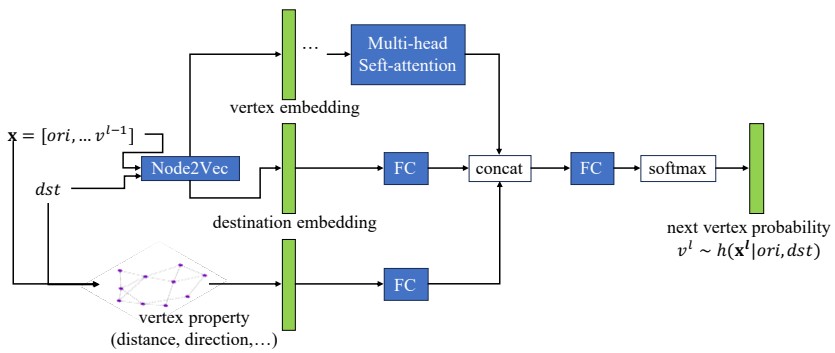

Figure 5: Our attention-based sequence model for planning.

---

**Algorithm 3:** Planning

**Input:** $nn_\theta$, $seq_\phi$, ori, dst
**Output:** A generated planned path $\mathbf{x}$.

1   $\mathbf{x} \leftarrow [ori]$
2   $window \leftarrow 1$
3   $idx \leftarrow window$
4   **while** $|\mathbf{x}| < MAX\_LEN$ **do**
5     $h_{seq} \leftarrow seq_\phi(ori, dst, \mathbf{x})$      // get evidence priori by sequence model
6     **if** $idx = window$ **then**
7       $\mathbf{x}_T \sim q(\mathbf{x}_T|\mathbf{x})$ by Eq.(8)           // diffuse prefix sequence
8       $\mathbf{w} \leftarrow \otimes_1^{window} \mathcal{U}[1, |V|]$      // random sequence with window size
9       $\mathbf{x}'_T \leftarrow \mathbf{x}_T || \mathbf{w}$     // concat prefix and random sampled sequence
10      $p_{nn} \leftarrow nn_\theta(\mathbf{x}'_T, T)[-window :]$   // get probabilities for later use
11      $idx \leftarrow 0$
12      $window \leftarrow window \times 2$     // increase window size exponentially
13     **end**
14     $\tilde{p}_{next} \leftarrow h_{seq} \odot p_{nn}[idx]$
15     $idx \leftarrow idx + 1$
16     $x_{next} \sim p_{next}$                   // sample next vertex
17     $\mathbf{x} \leftarrow \mathbf{x}.append(x_{next})$
18     **if** $x_{next} = dst$ **then break**;
19   **end**
20   **return** $\mathbf{x}$

---

## F   EXTENDED EXPERIMENTS

### F.1   IMPLEMENTATION DETAIL

**Dataset.** The raw datasets contains GPS trajectories from two cities, namely city A and B. We fetch the road network data from open street map [3]. The road network is a undirected graph. Then we apply a map matching algorithm Meert & Verbeke (2018) to bind the gps points of trajectories to the road network. Hence we convert the gps-trajectories into path data on graph. For city A/B, the graph of road network has 2717/2195 vertices. The number of paths are $118,535$/ $89,339$ with average length of 26.76 / 28.61.

**Parameter Setting.** For diffusion process, we set $T$ to 100 and the $\beta$ linearly increase from 0.0001 to 10. As for the training process, we choose the batch size as 16 and set the learning rate to 0.005. As for the Gaussian mixture model, we set 5 components. We train our model for 20

---

[3]https://www.openstreetmap.org

epochs on an Nvidia GeForce RTX 3090 with 24 GiB memory. Our source code can be found at `https://github.com/sdycodes/Graph-Diffusion-Planning`

**Evaluation Metrics.** NLL is defined as

$$NLL(\mathbf{x}, \theta) = - \sum_{j=1}^{|\mathbf{x}|-1} \log p_\theta(v_{j+1}|v_1, ..., v_j) \tag{18}$$

KLEV is calculated as below

$$KLEV(\mathcal{P}, \mathcal{P}') = D_{KL}(freq_{\forall(v_i, v_{i+1}) \in \mathbf{x}, \forall \mathbf{x} \in \mathcal{P}}(v_i, v_{i+1}) || freq_{\forall(v_i, v_{i+1}) \in \mathbf{x}, \forall \mathbf{x} \in \mathcal{P}'}(v_i, v_{i+1})) \tag{19}$$

where the $freq(\cdot)$ calculates all the visiting frequencies of all edges.

### F.2 PLANNING EXPERIMENT DETAIL

Table 4 provides a detailed information about performances. The unit of DTW is kilometer, which indicates the difference between planed paths and ground truth path. Smaller value indicates the planed path is more close to the ground truth path. For example, $2.434$ indicates that the algorithm planned paths averagely have a difference to the ground truth of $2.434$ kilometers. Our method can outperform the strongest baseline (*i.e.*, Navi) by 480 and 366 meters on City A and B respectively, indicating a perceivable difference in real road network.

Besides average performance, we also concern about the robustness of performances, *i.e.*, how steadily can we beat baselines? Table 5 provides the proportion of cases in which an algorithm can outperform Dijkstra's algorithm baseline. For instance, in city A, our GDP can beat Dijkstra's algorithm in 86.6% and 86.8% out of all test cases in terms of DTW and LCS, respectively. It achieves the best among all other methods.

Table 4: Evaluations for path planning. Less DTW or larger LCS means better performance.

| City | Metrics | Methods | | | | |
|------|---------|---------|-------|--------|--------|-------------|
|      |         | DA | NMLR | KS | Navi | GDP (Ours) |
| A    | DTW     | 2.434 | 9.109 | 2.405 | 2.164 | **1.683** |
|      | LCS     | 9.254 | 9.772 | 11.485 | 13.505 | **16.730** |
| B    | DTW     | 2.883 | 10.114 | 2.936 | 2.792 | **2.426** |
|      | LCS     | 9.233 | 10.066 | 12.207 | 14.390 | **16.856** |

Table 5: Proportion of test cases that can beat Dijkstra's baselines.

| City | Metrics | Methods | | | | |
|------|---------|---------|-------|------|------|-------------|
|      |         | DA | NMLR | KS | Navi | GDP (Ours) |
| A    | DTW     | — | 13.3 | 82.3 | 81.3 | **86.6** |
|      | LCS     | — | 70.3 | 83.8 | 79.0 | **86.8** |
| B    | DTW     | — | 11.1 | 76.9 | 78.7 | **81.3** |
|      | LCS     | — | 72.3 | 84.2 | 83.7 | **87.3** |

### F.3 ILLEGAL CASES DISCUSSION

**Hit ratio discussion.** We adopt the hit ratio (HR), which is the ratio between planned path hitting the destination to all paths. Intuitively the hit ratio is affected by the distance, so we split the sampled paths evenly into three groups based on their lengths and calculate the HR. Also, though our OD pairs is sampled from test data, we also shuffled the OD pairs to add the difficulty of planning. The HR comparison is illustrated in Figure 7. We found that for existing OD pairs from test data, the HR remains steadily around $98.6\%$ / $99.0\%$ in city A/B, respectively. As for shuffled OD pairs, the HR slightly drop to $94.2\%$ / $96.8\%$. This results indicates that our GDP ensures to reach the destination at a high success rate.

**Disconnection and loop discussion.** When we sample vertices directly based on the estimated distribution without help of post process like beam search. The model would generate illegal cases,

like disconnected vertices or loops. We switch off post process and calculate the proportion of illegal cases (see Table 6). Specifically, we randomly sampled 1000 OD pairs to plan paths. For each planned path, we calculate the ratio between number of disconnected vertices or loops to the length of planed paths. It can be shown that illegal cases happens very rarely (no more than 5% averagely in each path), which indicates that our model captures the graph structure.

Table 6: Proportion of illegal cases (mean $\pm$ std).

| Dataset | Disconnection | Loop |
|---|---|---|
| City A | $0.0479 \pm 0.0294$ | $0.0123 \pm 0.0788$ |
| City B | $0.0473 \pm 0.0309$ | $0.0203 + 0.107$ |

### F.4 COMPARISON BETWEEN GENERATED AND REAL PATHS

The generated and real paths on map are illustrated in figure 6. Some main roads are covered by more paths (red lines are more wide), which intuitively indicates generated paths follows real paths distributions.

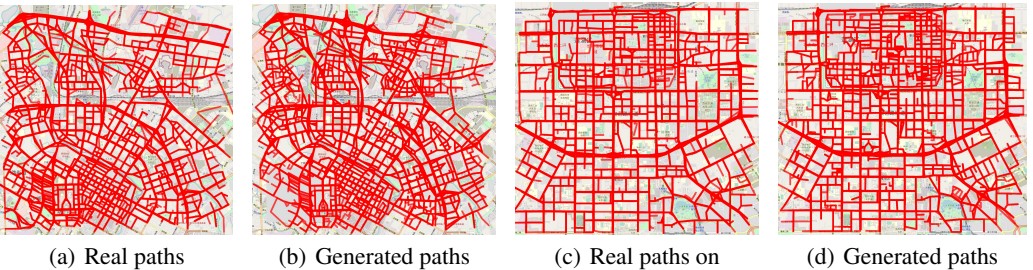

(a) Real paths      (b) Generated paths      (c) Real paths on      (d) Generated paths

Figure 6: Comparisons of real and generated paths on city A (Figure 6(a) and Figure 6(b)) and B (Figure 6(c) and Figure 6(d)).

Next we show our path lengths distributions to show the effectiveness of our Gaussian mixture model for path lengths. The Jensen-Shannon divergences of path length distributions are 0.0173, 0.0238 on City A and B, respectively. The histogram are illustrated in Figure 8. It can be shown that the length distribution of generated paths are very close to that of real paths, which validates the effectiveness of our Gaussian mixture model.

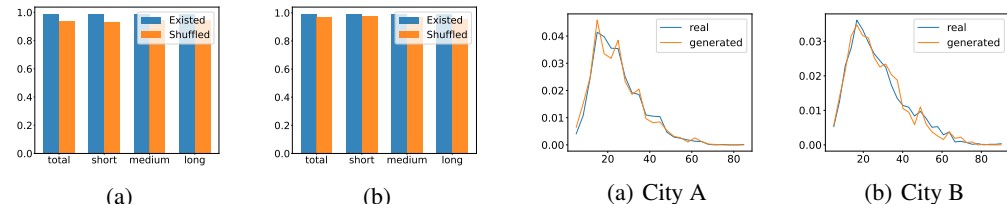

(a)      (b)      (a) City A      (b) City B

Figure 7: Hit Ratio (HR) of Planned paths on city A (left) and B (right)      Figure 8: Histograms of path lengths of City A and B.

### F.5 SCALABILITY AND EFFICIENCY

We random sample 1000 OD pairs from City A and B for execution efficiency validation with batch size 200 (see Table 7). It is not surprising that Dijkstra's Algorithm runs fastest. Other baselines such as NMLR and KS also run fast since they have a relatively simple neural network structure. Our method run faster than the strong baseline Navi. Considering Navi can support real-world application, the delay of our method is also acceptable.

Table 7: Execution time comparison in seconds (mean $\pm$ std).

| Dataset | DA | NMLR | KS | Navi | GDP (Ours) |
|---|---|---|---|---|---|
| City A | $0.0016 \pm 0.003$ | $0.024 \pm 0.003$ | $0.0187 \pm 0.0166$ | $0.347 \pm 0.205$ | $0.149 \pm 0.550$ |
| City B | $0.0015 \pm 0.002$ | $0.025 \pm 0.003$ | $0.0169 \pm 0.0176$ | $0.339 \pm 0.133$ | $0.112 \pm 0.315$ |

Next we discuss the scalability. In an Nvidia RTX 3090 with 24 GB memory, our method can support batch size up to 400 (See figure 9). We test time and memory consumption under different batch sizes from 10 to 400 on City B. As batch size increases, the memory consumption increases linearly, while the execution time for each planning task correspondingly reduces from $0.316$ to $0.115$ seconds.

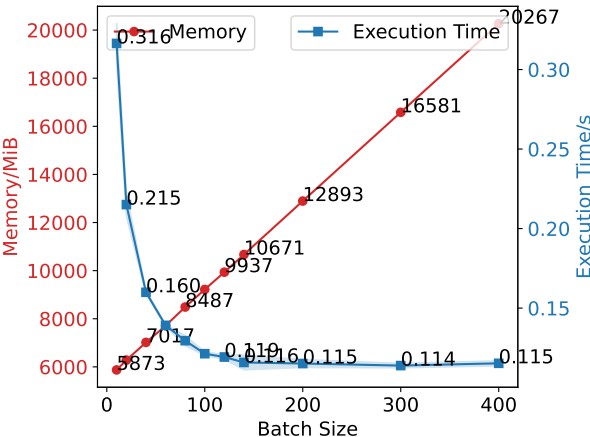

Figure 9: Memory and time consumption

