# OpenReview forum: "GRAPH-CONSTRAINED DIFFUSION FOR END-TO-END PATH PLANNING"
_ICLR.cc/2024/Conference — ICLR 2024 poster_

### Official Review · Reviewer_opqU · 2023-10-22

**Soundness:** 2 fair
**Presentation:** 4 excellent
**Contribution:** 3 good
**Rating:** 6
**Confidence:** 3

**Summary:**

This paper introduces a new technique for path planning, specifically path generation, on graphs such as road networks.

The key innovation is a diffusion model over a graph, capable of learning a probability distribution of paths from a set of expert demonstrations. Initially, a diffusion model for vertices is defined using the graph Laplacian, which is then extended to create a diffusion model for paths. Once learned, the model can be used for path generation through conditional sampling given a pair of origin and destination vertices. A significant advantage of this approach is its capacity to learn expert paths without assuming that they minimize linearly accumulative costs, a departure from standard search-based path planning.

The proposed method is evaluated on real-world road network datasets and is shown to outperform several existing methods.

**Strengths:**

**Originality**: Although previous work has leveraged diffusion models for path or motion planning, the use of diffusion models on a graph for path generation tasks seems novel. The ability of the proposed model to generate paths without assuming linearly accumulative costs is intriguing.

**Quality**: The overall quality of the work is high. The proposed method is well-designed and technically sound.

**Clarity**: The paper is well-written, with clear statements of the work's motivation and contribution.

**Significance**: The significance of the proposed method is evaluated on a real road network dataset, a strong point of this work, although I have some concerns about the dataset itself, as shown below.

**Weaknesses:**

While the proposed method's ability to bypass the linearly accumulative cost assumption is appealing, it's unclear if the path planning tasks for the collected datasets used in the experiment require such assumptions. In other words, I'm not sure if the dataset is fully suited to demonstrate the proposed method's significance.

In fact, the performance improvements over Dijkstra search are not very significant in terms of the DTW metric. With only average scores reported, it's unclear whether the proposed method demonstrates overall small improvements or if there are a few samples in the dataset where baselines completely failed to work. Similarly, the performance difference between the proposed method and CSSRNN is relatively small in Table 3. Does this suggest that path planning on the new dataset can be mostly solved by existing approaches that assume linearly accumulative costs?

Another concern is the computational cost. Classical path planners like Dijkstra are appealing because they can run quickly even on CPUs. In practical situations, people use path planning (e.g., route search on a map app) on mobile devices that don't always have sufficient GPU resources. It's not clear how much computation resources are required by the proposed method and other baselines.

**Questions:**

- Does the dataset contain a sufficient number of paths that don't follow the linearly accumulative cost assumption? At least the standard deviation or confidence intervals should also be reported in each table, but I wonder if the proposed method's strength could have been demonstrated more systematically using synthetic data that simulate paths that existing methods cannot plan or imitate from demonstrations? Such controlled experiments can be as important as real-world data evaluations.
- How does the proposed method compare to other baselines in terms of computational costs?

---

> ### Author Response · Authors · 2023-11-17
> **Response to W1-W3, Q1-Q2**
>
> Response to W1 and Q1:
>
> Thanks for your comments.
> Our dataset is a classic dataset for path planning in road network.
> In fact, our baselines NMLR and KS also choose this dataset for validation.
>
> Honestly, it is difficult to directly show that a sufficient number of paths that don't follow the linearly accumulative cost assumption.
> This is because many implicit goals are complex to model, and our data is insufficient for comprehensive verification.
> However, we notice that in experiment (Sec. 6.2) Dijkstra's algorithm underperforms.
> This underperformance suggests a considerable discrepancy in the DTW and LCS when comparing the algorithm's shortest paths with the ground truth paths, attributable to Dijkstra's algorithm's inherent focus on finding the shortest path.
>
> Response to W2:
>
> We are sorry for the confusion caused.
>
> The DTW value in the original manuscript is notably small because we calculate distances using the longitude and latitude coordinate system.
> A unit distance equates to approximately 110 kilometers.
> Consequently, a reduction of 0.001 signifies an improvement on the scale of hundred meters, which is significant in real-world planning.
> To address your concern, we have converted the distance unit to kilometers and altered the table contents to percentages for clarity.
> The original results are moved to App. F.2 of the revised manuscript.
>
>
> Regarding your concern about robustness, we present the proportion of scenarios in which each algorithm outperforms Dijkstra's algorithm.
> Our algorithm surpasses Dijkstra's algorithm in 86.6\% and 86.8\% of all test cases for DTW and LCS, respectively, marking the highest performance among all compared baselines (see App. F.2 of the revised manuscript for details).
>
>
> It is understandable that our improvement is modest compared to CSSRNN, as these recurrent neural network models are adept at capturing non-linear relationships in sequences.
> However, our primary focus is on planning (i.e., conditional sampling), not generation (i.e., unconditional sampling).
> CSSRNN cannot be applied to planning.
>
> Please refer to Sec 6.2 and App. F.2 of the revised manuscript.
>
> Response to W3 and Q2:
>
> Thanks for your comments.
> We are sorry for the confusion.
>
> Many modern navigation services are offered from the cloud, which spares users from the substantial storage and transmission costs associated with map data.
> Consequently, our method is not designed to run on users' local devices.
>
> To provide more clarity on our computation cost, we have included additional experiments.
> Overall, Dijkstra's algorithm is the fastest due to its simplicity.
> The execution time of our method is $0.112 \pm 0.315$ seconds, quicker than the baseline Navi, which has an execution time of $0.339 \pm 0.133$ seconds.
> Given that Navi is a real world navigation api developed by AMap (https://lbs.amap.com/), one of the leader location service provider in China, it can support real-world applications, the latency of our method is also acceptable for real-world applications.
> We conducted our experiments on an Nvidia RTX 3090 with 24 GB memory, capable of supporting up to a 400 batch size.
> As the batch size increases, the memory cost grows linearly, while the time consumption per origin-destination (OD) pair decreases from $0.32$ to $0.12$ seconds.
>
> We acknowledge that adapting our model to run on mobile devices is an intriguing challenge and consider it a subject for future work.
>
>
> For detailed information, please refer to App. F.5 of the revised manuscript.

---

> > ### Comment · Reviewer_opqU · 2023-11-20
> > **Thank you for the response**
> >
> > Thank you for the response! Although some of my initial concerns are resolved, I'm not yet convinced if the current experimental result validates the ability of proposed method to handle non-linearly accumulative costs.
> >
> > Specifically, I found that just outperforming Dijkstra is not necessarily mean that the proposed method can plan a path based on non-linearly accumulative costs. Suppose a problem where there are two roads from the start to the goal; one road is longer and safer, and the other is shorter but not safer. Safe drivers may choose the longer-safer roads, which cannot be imitated by Dijkstra search "if the edge cost is given solely by its length" (i.e., shorter road will be planned). But if the cost is instead given based on safety metrics, Dijkstra search can find a longer-safer choice as the lowest-cost solution. So the problem here is actually about how the underlying cost function is defined, rather than if the cost had been linearly accumulated in demonstration paths or not. Keeping this in mind, it is not obvious whether the current dataset involves demonstration paths that are really based on non-linearly accumulative costs or paths that are just based on unknown cost functions that are still able to be linearly accumulative. Please correct if I'm misunderstanding something.
> >
> > I just wonder if it's possible to just remove the claim that the proposed method can account for non-linear accumulative costs. Although it could decrease the significance of the work, but doing so would at least avoid arguing the contributions that are not empirically supported.

---

> ### Author Response · Authors · 2023-11-20
>
> We are sorry for the confusion.
> Our main contribution is the diffusion process on graph space.
> Leveraging this, we are able to accomplish path planning via conditional sampling (ie. end-to-end planning).
>
> The main difference between our end-to-end planning algorithm and current data-driven approach is that we avoid the searching framework.
> Traditional searching frameworks necessitate:
>
> i) explicit modeling cost for each edge.
>
> ii) finding the path with smallest summation of cost of edges.
>
> These prerequisites are associated with two major limitations of the search-based framework:
>
> i) Given that each edge's cost must be modeled, all planning objectives and their interconnections need to be explicit.
>
> ii) As the total path cost is a linear sum of the individual edge costs, the method is inherently bound to linear accumulation.
>
> Our model sidesteps the search-based framework by utilizing conditional sampling.
> The associated distribution is modeled by a U-net framework with convolutional and fully-connected strcuture.
> So it is reasonable that our model is capable of capturing non-linear and high order dependent between vertices and edges within a path.
> However, your concern regarding empirical testing is well-noted. We have restated non-linear accumulation as "another possible reason", "provide potential benefits for planning" to make the statement more rigorous. Please see Section 1 for additional details.

---

> > ### Comment · Reviewer_opqU · 2023-11-20
> > **Thank you**
> >
> > Sure. I have confirmed the response and revision. Thank you for the clarification!

---

### Official Review · Reviewer_GtR8 · 2023-10-29

**Soundness:** 4 excellent
**Presentation:** 4 excellent
**Contribution:** 3 good
**Rating:** 6
**Confidence:** 2

**Summary:**

This paper proposes a diffusion-based model for end-to-end data-driven path planning, called GDP. GDP models path planning as a conditional sampling task. Its objective is to determine the probability distribution of paths given an origin and destination. To solve this task, GDP uses a diffusion-based architecture.

The authors evaluated the GDP model using two real datasets, City A and City B. They compared against traditional optimization methods as their baselines, including Dijkstra's algorithm, NMLR, Key Segment, and Navigation API from Amap. The result shows that the paths generated by GDP are closer to the ground-truth paths than those baselines.

**Strengths:**

* The paper proposes a sound diffusion-based model for path planning. The proposed model achieves better performance than the traditional methods on public datasets.

* The paper is very well-written. The structure is clear, and the evaluation is thorough.

**Weaknesses:**

* The improvement from the GDP model over the Navi baseline is relatively small.

**Questions:**

N/A

---

> ### Author Response · Authors · 2023-11-17
> **Response to W1**
>
> Response to W1:
>
> Thanks for your question!
> We apologize for any confusion caused.
>
> It seems the aspect that most concerns you is the Dynamic Time Warping (DTW) comparison, particularly because the numerical values are very small, ranging from 0.023 to 0.021.
> This is because we calculate distances using longitude and latitude coordinates.
> In such calculations, a unit distance corresponds to approximately 110 kilometers.
> Therefore, a reduction of 0.001 represents a significant improvement on a hundred-meter level, which is quite perceptible in real-world route planning.
> To address your concern, we have revised the distance unit to kilometers and modified the table to display improvement percentages for convenience. The original results can now be found in App.F.2.
> Please refer to Sec 6.2 of the revised manuscript for these updates.
>
> Additionally, as introduced in Sec. 6.1 of the original manuscript, the Navi baseline uses the planning API from Amap (also known as Gaode Map), one of the largest mobile digital map providers in China, boasting over 100 million daily active users.
> Amap has invested considerable resources in enhancing navigation intelligence by developing a complex framework that integrates both rule-based and data-driven approaches, making it an exceptionally strong baseline for comparison.

---

> > ### Comment · Reviewer_GtR8 · 2023-11-19
> > **Thank you for your response**
> >
> > Thank you for your response. I will keep my score.

---

### Official Review · Reviewer_j9CR · 2023-10-30

**Soundness:** 3 good
**Presentation:** 3 good
**Contribution:** 3 good
**Rating:** 8
**Confidence:** 3

**Summary:**

This work propose a GDP, a diffusion model on graphs, which is able to conduct path planning in an end-to-end manner. The experiments are conducted on two real city datasets and compared against four baseline planners, showing that GDP is able to generate paths very close to the groundtruth.

**Strengths:**

1. The technical part is solid and using diffusion model to address the path planning is interesting.
2. The performance is evaluated on big and real city datasets and promising results are shown.

**Weaknesses:**

I can not accurately tell the weakness of this paper. Please see my questions below.

**Questions:**

1. How do you deal with the unsuccessful planning tasks?
2. Have the authors compared the latency of GDP with other models, because latency is also crucial in giving real-time path solution, and I am concerned that diffusion model can be slow due to many iterations of run.
3. What is the motivation for unconditional path generation? Is it for preparing a high-quality roadmap for the following specific tasks?
4. How do you compare your method/contributions with Motion Planning Diffusion [R]?

I hope the authors can address my questions and I will be glad to adjust the score afterwards.

[R] Carvalho, Joao, et al. "Motion planning diffusion: Learning and planning of robot motions with diffusion models." arXiv preprint arXiv:2308.01557 (2023).

---

> ### Author Response · Authors · 2023-11-17
> **Response to Q1-Q4**
>
> Response to Q1:
>
> Thanks for your question. We are sorry for the confusion.
> In fact we can choose any search-based planning algorithm as our fallback strategy (eg. the A* algorithm) to guarantee that our method always hits the goal.
> This will not significantly impact our performance, as our method maintains a high hit ratio of 98.6\%, as illustrated in App. F.3 of the original manuscript.
>
> Response to Q2:
>
> Thanks for your question.
>
> We have added an experiment on execution efficiency in App. F.5 of the revised manuscript.
> Dijkstra's algorithm is the fastest due to its simplicity.
> The execution time of our method is $0.112 \pm 0.315$ seconds, which is faster than the baseline Navi ($0.339 \pm 0.133$ seconds).
> Given that Navi is a real world navigation API developed by AMap (https://lbs.amap.com/), one of the leader location service providers in China, it can support real-world applications, the latency of our method is also acceptable for real-world applications.
> Our experiments were conducted on an Nvidia RTX 3090 with 24 GB of memory, accommodating up to a batch size of $400$.
> As the batch size increases, the memory cost rises linearly, but the time consumption per OD pair decreases from $0.32$ to $0.12$ seconds.
>
> Please refer to App. F.5 of the revised manuscript for detailed information.
>
> Response to Q3:
>
> Thanks for your question.
>
> Unconditional path generation is one of the solutions for trajectory data mining [1], serving as a building block for various intelligent transportation applications, such as road network planning, traffic light scheduling, and traffic bottleneck detection [2].
> Recently, the success of generative models in image and text generation tasks has spurred interest in developing models capable of generating paths or trajectories, making it a compelling research topic [3].
> Researchers hope that the capabilities of generative models will enhance the mining of implicit patterns from path data, enabling the creation of smarter transportation systems.
>
> However, these methods are not suited for high-quality road map generation, primarily because they require the road network (i.e., graph structure) as an input.
>
> [1] Yu Zheng. Trajectory data mining: an overview. ACM Transactions on Intelligent Systems and Technology (TIST), 2015.
>
> [2] Zheng Y, Capra L, Wolfson O, et al. Urban computing: concepts, methodologies, and applications[J]. ACM Transactions on Intelligent Systems and Technology (TIST), 2014, 5(3): 1-55.
>
> [3] Generative Adversarial Networks for Spatio-temporal Data: A Survey
>
> Response to Q4:
>
> Thanks for your question.
> The paper you mentioned is indeed interesting.
>
> The primary distinction between our method and that work lies in the space where the diffusion process occurs: our method operates in graph space, while the other work functions in high-dimensional Euclidean space.
> As mentioned in Sec. 1 of our original manuscript, this difference renders the adoption of a Gaussian-based diffusion model formidable and necessitates the development of a new diffusion process.
> To address this, we drew inspiration from the heat conduction process on graphs and devised a novel diffusion model capable of diffusing distribution across graph space, thereby enabling us to capture the graph structure effectively.
>
> We have added discussion of this paper into Related work, please refer to Sec. 7 of the revised manuscript.

---

> > ### Comment · Reviewer_j9CR · 2023-11-20
> > **Thank you for the response**
> >
> > Thank you for addressing most of my concerns, I have increased my score.

---

### Official Review · Reviewer_nmfg · 2023-10-31

**Soundness:** 3 good
**Presentation:** 3 good
**Contribution:** 3 good
**Rating:** 6
**Confidence:** 3

**Summary:**

The paper tackled the problem of building a novel conditioned sampling method to mimic user historical paths on a graph. To implement the sampling process in the manner of diffusion models, the authors reviewed requirements based on previous work. They formulated the forward and backward process following the heat conduction on the graph. Experimental comparisons with the proposed method and existing methods show the promising performance of the diffusion-based method for the task.

**Strengths:**

- Clear explanations of the motivation and background concept (of conditioned samples through diffusion models) for the targeting task (i.e., end-to-end path planning).
- Mathematically solid contribution through the heat conduction on graphs to build diffusion models.
- Good experimental performance for the end-to-end path planning.

**Weaknesses:**

- Following the existing literature (e.g., Austin et al. 2021 and Yi et al. 2023), the novelty of the contribution is less explained (the background idea and reasons to follow the heat conduction on graphs, although the performance is good).
- Possibly, the discussion between the continuous space (i.e., latitude-longitude vectors) and discrete space (i.e., the sequence of nodes on a graph) is not included in the paper (e.g., NeurIPS'23 DiffTraj.); the difficulty of discrete spaces or similarity between the two models are better to be explained.
- Many details are included in the appendix, making the main paper hard to read and follow the details.

**Questions:**

- Do we have any discussions on the parameters of diffusion processes: examples are the length of diffusion time (i.e., t=100 in Fig. 3 in city B).
- Related to the 2nd point of the weakness above, I'm curious about the relation between the diffusion models among those for continuous spaces and those for discrete spaces (i.e., this paper). I found that DiffTraj in NeurIPS'23 tacked a similar problem, but they seem to focus on the continuous space. Therefore, please clarify the difficulty of discrete spaces or the similarity between the two models. (Of course, if such a paper overlaps with the submission of ICLR, you can refer to other papers used as baselines of DiffTraj). To clarify the difference, your contributions are expected to be clarified and explained well.
- I cannot completely follow the discussion of introducing U-net for the purpose of $\mathbf{x}_0$: Could you give some additional explanations? (In experiments, for example, they are static (OD-pairs seem to be known), but some probabilistic characterization as a distribution p(x0) is required for the diffusion process; is this right? In Line 2 of Alg1, $\mathbf{x}_0$ is sampled from $p(\mathbf{x}_0)$, but I'm confused that $\mathbf{x}_0$ is already known?)
- As the dataset contains multiple OD pairs, I’m curious about the std. of the metrics (DTW, LCS): Does the GDP show good performance for almost all OD pairs? Are there any specifically difficult trajectories according to their conditions?

**Details Of Ethics Concerns:**

Location data often have privacy concerns, but it depends on the quality of the input of any methods to replicate trajectories. As far as I read the paper, the authors seem to take care of them.

---

> ### Author Response · Authors · 2023-11-17
> **Response to W1-W3, Q1-Q2**
>
> Response to W1:
>
> Thanks for your comment.  We are sorry for the confusion.
>
> In the beginning of Sec. 4 and Sec. 4.1 of the original manuscript, we have highlighted the idea or reasons to follow the heat conduction.
> Specifically, our goal is to design a diffusion model for categorical values (i.e., vertices) that can effectively capture the graph structure, making it more suitable for graph space tasks such as path planning.
> While existing literature has explored diffusion models for categorical values, these studies do not focus on designing a process that captures the graph structure.
> However, recent research has highlighted the potential of tailoring diffusion processes to different categorical values.
> For instance, Austin et al. in their paper stated,"We argue that for most real-world discrete data, including images and text, it makes sense to add domain-dependent structure to the transition matrices $Q_t$ as a way of controlling the forward corruption process and the learnable reverse denoising process."
>
> To further clarify our motivation, we have made modifications to Sec. 1 in the revised manuscript.
>
> Response to W2 and Q2:
>
> Thanks for your comments and we are sorry for making you confused.
>
> The primary distinction between these two approaches stems from specific task requirements.
> For path planning in a road network, the output must be a connected path on the graph.
> This is because vital metadata such as the number of lanes, traffic lights, traffic signs, and speed limits are typically organized based on road segments rather than longitudinal or latitudinal coordinates [1].
> As a result, a path on a graph is more suitable for the post-processing of navigation applications.
> Conversely, for certain trajectory analysis applications, the focus is more on understanding the relationships among spatial locations and areas.
> In these cases, generating coordinates is more practical.
>
> Moreover, since the road network graph also includes longitude and latitude information for vertices, converting connected paths into sequences of points with these coordinates is straightforward.
> However, models that generate coordinate sequences face challenges in converting these into paths on a graph.
> This difficulty arises because the coordinate sequence generation process does not account for structural constraints.
> Such models may need to use map-matching techniques [2] to align the coordinate sequence with the road network.
>
> These differences and their implications are further discussed and clarified in Sec. 7 of the revised manuscript.
>
> [1] Zheng Wang, Kun Fu, Jieping Ye. Learning to Estimate the Travel Time. KDD 2018: 858-866
>
> [2] Paul Newson, John Krumm. Hidden Markov map matching through noise and sparseness. GIS 2009: 336-343
>
> Response to W3 and Q1:
>
> Yes. In Sec. 6.1 of the original manuscript, we explained our experiment detail and put other settings in App. F.1.
> We are sorry for making you confused.
> Due to the page limit, we have to put some details in the appendix.
> We have thoroughly revised the paper to ensure that all detailed information provided in the appendix is appropriately referenced in the main content.

---

> ### Author Response · Authors · 2023-11-17
> **Response to Q3, Q4**
>
> Response to Q3:
>
> We are sorry for the confusion.
>
> The Algorithm 1 is designed for unconditional path generation, so it does not require any OD pairs' information.
>
> The line 2 of Algorithm 1 simply denotes sampling raw path data from dataset under the real data distribution.
> The $\mathbf{x}_0$ represents a raw path data, which contains a sequence of one-hot vectors.
> Each represents a vertex of a path.
>
> The U-net structure is worked as $nn_\theta$ in algorithms and equations, it takes the diffused path (during training) or random sampled vertex sequences (during evaluating) as input and outputs denoised sequences as generated path.
> It only requires the sequences and diffusion step $t$ as input.
>
> We have revised the end of Sec. 4 and Algorithm 1 of the revised manuscript for better understanding.
>
> Response to Q4:
>
> Thanks for your comments.
> We are sorry for the confusion.
>
> Regarding your concern about robustness, we present the proportion of scenarios in which each algorithm outperforms Dijkstra's algorithm.
> Our algorithm surpasses Dijkstra's algorithm in 86.6\% and 86.8\% of all test cases for DTW and LCS, respectively, marking the highest performance among all compared baselines (see App. F.2 of the revised manuscript for details).
>
> It is not suitable to use mean $\pm$ std for the performance robustness, since the trajectory length are range from 10 vertices to hundreds of vertices (As shown in Fig. 8 of the original manuscript).
> So it is understandable the performances will have a large variance.
>
> Please refer to Sec 6.2 and App. F.2 of the revised manuscript.

---

> > ### Comment · Reviewer_nmfg · 2023-11-20
> > **Thank you for your responses**
> >
> > I appreciate all the comments and updates clarifying some concerns and my understanding from the author. Detailed comments on experiments (i.e., DTW/LCS, large variances in results of trajectories) and Alg 1 helped me understand the contribution more clearly.

---

### Official Review · Reviewer_YKA1 · 2023-11-01

**Soundness:** 3 good
**Presentation:** 3 good
**Contribution:** 4 excellent
**Rating:** 6
**Confidence:** 4

**Summary:**

The authors consider the problem of sampling paths on a graph. Their approach makes use of recent developments on argmax flows, that allow for the definition of diffusion with categorical variables.
Their insight is that they can use a heat equation to model the transition probability matrix of a specific graph, with the heat transfer defined by the adjacency, balanced by the degree.
They then use this to define their forward diffusion process. They use a sequence of conditionally independent nodes to model a path, and use post processing to ensure connectivity. They demonstrate the value of their approach on a series of datasets, outperforming baselines.

**Strengths:**

- Algorithms on graphs are very useful in practical applications
 - The heat conduction approach is elegant, and I think quite clever

**Weaknesses:**

- The work seems to cut some corners when it comes to sampling connected paths, post-hoc processing is required
 - The path length seems like an integral part of the problem, but is only discussed very briefly in the appendix.

**Questions:**

Overall I enjoyed reading this paper. I think the core of the method is good, interesting, and useful. I do have some questions:

 - In Table 3, the authors present an ablation as "trivial uniform diffusion". Do the authors mean the "generic categorical diffusion" with Q = $\alpha_t I + \beta_t 1 1^T / V$? If this is the case, the authors should make that more clear so that readers may enjoy the significant improvement their work brings. If this is not the case, and the authors mean something else, they should run another ablation using "generic categorical diffusion" as a baseline, since that seems like the relevant benchmark to beat.
 - Can the authors elaborate on the beam search aspect, and how often a sampled path is actually invalid (i.e., disconnected, or with loops, etc), and what heuristics are used to fix those sampled paths?
 - Can the authors discuss their Guassian mixture model for path lengths in some more detail? While some metrics are provided in the appendix, it is not clear what information this model takes, if any. It seems to me that path length is highly dependent on origin and destination, so I assume their mixture model takes these as input. Can the authors elaborate exactly what the structure of this model is?
 - What is the scaling of the algorithm? Especially compared to Dijkstra? Both in terms of path length, and number of graph vertices.
 - Can the authors elaborate on how they generalize to paths without learning a joint distribution across time? Sampling nodes independently certainly would not yield a sensible path. I could imagine this would work if the reverse process was conditionally independent (i.e. $x^i_{t-1} | x_t$, the latter $x_t$ being _all_ nodes instead of $x_t^i$), but eqn 9 does not actually seem to suggest that. The text touches on this (end of paragraph 3 in 4.3, "masking $\hat{x_0}$"), but to me this statement is quite uninformative, and seems like a very essential point of the paper.
 - Does it generalize to different graphs? i.e. do you need to retrain for each city?

Minor points:
The text is generally well written, but it seems like a few paragraphs were missed during the proof reading stage.
A non-exhaustive list:
 - first paragraph of 4.2
 - first paragraph of sec 5
 - App. D
 - 6.2 paragraph 2
 - typo in Fig 2. "Pais", it is also confusing for 3rd panel to say "Real", since the text says these were generated using Navi

---

> ### Author Response · Authors · 2023-11-17
> **Response to W1, W2, Q1-Q4**
>
> Response to W1 and Q2:
>
> We apologize for any confusion caused.
> We have explained how beam search is used to maintain connectivity in App.C.2 of the revised manuscript.
> Specifically, the beam search is conducted after the reverse process when we have obtained the final estimated distribution $\mathbf{\hat{x}}_0$.
> Ideally, beam search operates in a breadth-first search manner.
> We start by sampling $n$ vertices from the distribution of $\mathbf{\hat{x}}_0^0$ (with replacement).
> Subsequently, we sample (up to $n$) neighbors of each vertex from the distribution $\mathbf{\hat{x}}_0^1$.
> This process continues, and we always keep (up to) $n$ sequences with the highest probability in a queue.
> Finally, we sample from these $n$ sequences to get the final path.
> In practice, however, setting $n$ to 1 suffices for good performance and efficient execution.
>
> Without beam search, path legality is not guaranteed.
> If our model fails to generate an ``almost legal'' path without beam search, it suggests that the model does not accurately capture the path pattern.
> To address this concern, we discuss illegal cases (i.e., disconnections and loops) in App. F.3 of the revised manuscript, alongside the original hit ratio discussion.
> In summary, our model maintains a low ratio (less than 5%) of illegal vertices in each path, indicating its proficiency in capturing the path pattern.
>
> Response to W2 and Q3:
>
> We are sorry for the confusion.
> The Gaussian mixture model is detailed in the appendix primarily because it is utilized for unconditional path generation.
> In other words, the Gaussian mixture model takes only the path length as input, without additional information such as origin and destination.
> In practice, a Gaussian mixture model with four components effectively models the path length distribution.
>
> The process of unconditional path generation can be summarized as follows:
> i) Sample from the Gaussian mixture model to determine the path length (denoted as $l$);
> ii) Generate a random vertex sequence of length $l$;
> iii) Perform the reverse process to denoise the random vertex sequence, obtaining the estimated distribution of $\mathbf{\hat{x}}_0$; and
> iv) Apply beam search to ensure connectivity.
>
> For conditional path generation, we employ an exponentially increasing window size strategy, negating the need for a Gaussian mixture model for a predefined path length.
> We adopt this approach instead of directly sampling a long random sequence because we hope a progressive manner will better inform the network $nn_\theta$ about the path prefix, which results in more effective planning.
> The detailed process is outlined in Algorithm 3 in App.E. of the original manuscript.
> It begins with the origin as the initial path sequence, which is then diffused and concatenated with one randomly sampled vertex, yielding a (diffused) path of length two.
> This path is then denoised through a reverse process.
> Next, the path sequence is diffused again and concatenated with two new randomly sampled vertices, followed by another reverse process.
> This results in a path of length four.
> This iterative process continues until the goal is reached or the maximum length limit is hit.
> The exponential increasing strategy eliminates the need to predefine the path length before planning and enhances efficiency by avoiding a one-by-one diffusion process.
>
> Response to Q1:
>
> We are sorry for the confusion.
> Yes, the trivial uniform diffusion means ``generic categorical diffusion''.
> We have modified the description for unification in Sec. 6 of the revised manuscript.
>
> Response to Q4:
>
> Thanks for your comments.
> We have added an experiment on execution efficiency in App. F.5 of the revised manuscript.
> Dijkstra's algorithm is the fastest due to its simplicity.
> The execution time of our method is $0.112 \pm 0.315$ seconds, which is faster than the baseline Navi ($0.339 \pm 0.133$ seconds).
> Given that Navi is a real world navigation API developed by AMap (https://lbs.amap.com/), one of the leader location service providers in China, it can support real-world applications, the latency of our method is also acceptable for real-world applications.
> Our experiments were conducted on an Nvidia RTX 3090 with 24 GB of memory, accommodating up to a batch size of $400$.
> As the batch size increases, the memory cost rises linearly, but the time consumption per OD pair decreases from $0.32$ to $0.12$ seconds.
>
> Please refer to App. F.5 of the revised manuscript for detailed information.

---

> ### Author Response · Authors · 2023-11-17
> **Response to Q5, Q6, Q7**
>
> Response to Q5:
>
> Thanks for your comments.
> We apologize for any confusion caused.
>
> The forward process operates independently.
> However, in the reverse process, the estimation of $\mathbf{\hat{x_0}}$ is not independent because the network $nn_\theta$ adopts a U-net structure.
> This structure includes convolutional and fully connected layers, causing the outputs of each vertex to be interdependent.
> As a result, each $\mathbf{\hat{x}}_0^{i}$ is not independent.
> The ``mask'' you mentioned is used to prevent the sampling of unconnected vertices; it refers to the beam search.
>
> We have revised the end of Sec. 4 of the revised manuscript for better understanding.
>
> Response to Q6:
>
> Thanks for your comments.
> Yes, we need to retrain for each city.
> We think the generalization is an interesting problem and consider it as future work.
> Also, we think that retrain for each city is acceptable.
> If we hope to get a model that can genralize to orher cities, we may need to train the model with various cities' data with a larger model, which would incur larger computation cost.
>
> Response to Q7:
>
> We have revised the typos accordingly.

---

> > ### Comment · Reviewer_YKA1 · 2023-11-23
> >
> > Thank you for addressing my comments and concerns.
> >
> > There is one issue that I think might be better addressed in the paper. Although I do not think it is reason to reject the paper, it could be added to the discussion. This issue is that the authors algorithm for determining length is "greedy", i.e. it stops when an allowable path has been sampled. While this is allowable, I think that path and path length are highly correlated (i.e. a short path would probably go over highways, while a longer route would probably take more scenic roads). This means that the sampling scheme the authors have devised would preferentially yield short segments, which is in contradiction with the justification the authors provide with in the introduction.
> >
> > There is another small concern remaining, which is that my concerns regarding writing clarity have not been addressed in the current manuscript, the following sections should be carefully proofread since there are a lot of grammatical mistakes (missing articles, wrong conjugations, etc). Also not a reason to reject, but I think the authors should address it:
> >  - first paragraph of 4.2
> >  - first paragraph of sec 5
> >  - App. D
> >  - 6.2 paragraph 2
> >
> > That said, I think the things that concerned me most have been addressed, I am in particular happy to have heard the authors clarify that they indeed beat generic categorical diffusion, which I think really demonstrates the value of the paper.

---

> > > ### Author Response · Authors · 2023-11-23
> > >
> > > Thanks for your response.
> > >
> > > For your concern about the search manner issue, we really appreciate that.
> > > We think terminating the algorithm right after it reaches the destination can improve efficiency.
> > > It might makes the sampling distribution slightly different from original proposal provieded by our diffusion model.
> > > Experiments in Sec. 6.2 shows that even though our algorithm still performs better.
> > > We think designing an algorithm free from any post process is interesting.
> > > We have added this discussion in Sec. 8 of the revised manuscript.
> > >
> > > We are sorry for those grammar errors.
> > > We did not understand you at first.
> > > Now, we have revised the paragraphs you mentioned.
> > > We will thoroughly check the whole paper for the camera-ready version if accepted.

---

### Author Response · Authors · 2023-11-17
**Thanks for all reviewers.**

We sincerely appreciate you for your insightful comments and helpful suggestions on our paper.
We have given careful consideration to the suggestions made and strengthened the paper by addressing all the comments as below.
All changes in the revised manuscript has been highlighted in blue for your convenience.

---

### Public Comment · ~Dingyuan_Shi1 · 2024-05-27
**Source code link change to: https://github.com/dingyuan-shi/Graph-Diffusion-Planning**

Source code link change to: https://github.com/dingyuan-shi/Graph-Diffusion-Planning

---

### Meta-Review · Area_Chair_2E58 · 2023-12-06

**Metareview:**

This paper proposes a method for planning paths in graphs using diffusion. Whilst diffusion has been previously used for planning, the specific application to path planning on a graph has some novelty and presents unique challenges. All reviewers acknowledged the method is sound and adequately evaluated.

**Justification For Why Not Higher Score:**

+ Paper is focused on a very specific problem, limiting the breadth of its appeal
+ Reviewers commented on the soundness of the approach, but raised few other strengths of the work

**Justification For Why Not Lower Score:**

+ All reviewers agree the paper makes a publishable contribution

---

### Decision · Program_Chairs · 2024-01-16

Accept (poster)